# Preparation of Nanochitin from Crickets and Comparison with That from Crab Shells

Kana Kishida [1], Toshifumi Mizuta [2], Hironori Izawa [1,3,4] and Shinsuke Ifuku [1,3,*]

1    Graduate School of Engineering, Tottori University, 4-101 Koyama-Minami, Tottori 680-8550, Japan
2    Technical Department, Tottori University, Tottori 680-8550, Japan
3    Center for Research on Green Sustainable Chemistry, Tottori University, Tottori 680-8550, Japan
4    Faculty of Engineering, University of Miyazaki, 1-1 Gakuen Kibanadai-Nishi, Miyazaki 889-2192, Japan
*    Correspondence: sifuku@tottori-u.ac.jp

**Abstract:** Crickets are gaining worldwide attention as a nutrient source with a low environmental impact. We considered crickets as a new source of chitin raw material. Chitin isolated from crickets was successfully converted to nanochitin by pulverization. First, chitin was obtained from cricket powder in a 2.6% yield through a series of chemical treatments. Chitin identification was confirmed by FT-IR and $^{13}$C NMR. The chitin had an α-type crystal structure and a deacetylation degree of 12%. Next, it was pulverized in a disk mill to obtain nanochitin. Cricket nanochitin was of a whisker shape, with an average fiber width of 10.1 nm. It was larger than that of crab shells, while the hydrodynamic diameter and crystal size were smaller. Such differences in shape affected the physical properties of the dispersion. The transmittance was higher than that of crab nanochitin due to the size effect, and the viscosity was smaller. Moreover, the dry non-woven cricket nanochitin sheets were more densely packed, and their modulus and breaking strength were greater.

**Keywords:** nanochitin; cricket; nanowhisker; reinforcing material; nanocomposite

## 1. Introduction

Chitin is a polysaccharide with *N*-acetylglucosamine as its building block. It is the main component of the outer skin (cuticle) that covers the bodies of arthropods such as crustaceans and insects. It is the second most abundant biomass resource after cellulose [1]. However, while some chitin is used as an intermediate for chitosan and glucosamine, the uses of chitin itself are few. Therefore, it has been called the "last biomass" left behind that can be used on a large scale in the future. This is due to low solubility and dispersibility in solvents. Its characteristics make it difficult to formulate and process, making it difficult to commercialize. It also makes it difficult to verify the function. Chitin derived from crab shells was converted into nanochitin via mechanical pulverization by applying nanocellulose production technology [2]. Nanochitin is a fibrous substance with a width of about 10 nm and can be stably dispersed in water, allowing it to be mixed with other ingredients and processed into various forms [3]. In vitro and in vivo studies have also revealed various physiological functions. They include health care effects by application to the skin [4,5] or ingestion [6,7], and can also have effects when applied to plants [8,9]. Other characteristics of nanochitin include excellent mechanical properties, i.e., high Young's modulus and fracture strength. Nanochitin is composed of microcrystals, and its structure consists of chitin molecules in rigid extended chains that are regularly arranged with intermolecular hydrogen bonds. Therefore, nanochitin has high strength, high elasticity, and low thermal expansion [10]. Consequently, it can be used as a nanofiller to strengthen materials. Resins containing nanochitin were created to significantly improve their mechanical properties [11,12]. In addition, nanochitin is less likely to cause optical scattering due to its size effect, a property that is less likely to affect the transparency of the material. Taking advantage of a series of biological characteristics, cosmetics, health foods,

and health care agents for animals containing nano-chitin have been commercialized. The use of nanochitin is expected to expand. However, crab catches have been declining in Japan in recent years, and if demand for these crabs increases, there may be a shortage of crab shells in the future. Therefore, there is a need to search for new chitin sources that can replace crab shells.

Demand for protein is increasing due to global population growth and economic development. The Food and Agriculture Organization of the United Nations reports that traditional livestock-dependent protein production will no longer be able to meet demand by 2025 to 2030. To meet that challenge, it recommends an insect diet as an alternative to animal protein [13]. Insects have long been consumed in Asian and African regions because of their high nutritional value, being rich in lipids, proteins, and minerals. One advantage of the insect diet is its low environmental impact [14,15]. For example, crickets, a typical edible insect, require 2.1 kg of feed and 420 L of water to produce 1 kg of edible parts. Cattle, on the other hand, require 25 kg and 22,000 L, respectively. Since chitin is the main component of cricket cuticles, it could be used as a new source of nanochitin. To date, there are no reports on the production and detailed characterization of cricket-derived nanochitin. Therefore, in this study, nanochitin was produced from crickets. We then evaluated the differences in morphology and physical properties to conventional nanochitin derived from crab shells and verified its potential as a raw material for composite materials and other applications.

## 2. Materials and Methods

### 2.1. Materials

House cricket (*Acheta domesticus*) powder was purchased from Futurenaut (Takasaki, Japan). It was raised in a Good Manufacturing Practice (GMP)-certified indoor facility in Thailand. $\alpha$-Chitin powder from crab shells was acquired from Koyo Chemicals Industry Co., Ltd. (chitin TC-L, Sakaiminato, Japan). The degree of deacetylation of chitin was 6%. Sodium hydroxide, oxalic acid, and organic solvents with analytic grade were mainly purchased from Fujifilm Wako Chemicals Ltd. and used as received.

### 2.2. Preparation of Chitin from Cricket Powder

According to the scheme in Figure 1 [16], Chitin was isolated from house crickets. Dry cricket powder (100 g) was defatted in a Soxhlet extractor with 4.2 L of a mixture of toluene and ethanol (2:1 (*v*/*v*)) for five days. After air-drying, proteins were removed by treatment with 1 M NaOH solution at 95 °C with stirring for at least 6 h. The process was repeated until no protein was detected by quantifying protein concentration by the Bradford method (Quick Start, BioRad, Hercules, CA, USA) [17]. After that, the solid residue was collected by suction filtration using a 90 mm diameter quantitative filter paper and washed with distilled water until the filtrate turned to neutral pH. The inorganic minerals from the crickets were then treated with a 1% oxalic acid solution at room temperature for at least 3 h with stirring to remove the minerals. This process was repeated until thermogravimetric analysis detected no mineral content (Thermo plus EVO II, Rigaku, Tokyo, Japan). The sample was then filtrated and washed with water. After demineralization, it was treated with 1% sodium hypochlorite for decolorization for at least 3 h at room temperature with stirring, and then filtered, and washed with water. After a series of extraction processes, the yield was determined from the dry weight of the purified product.

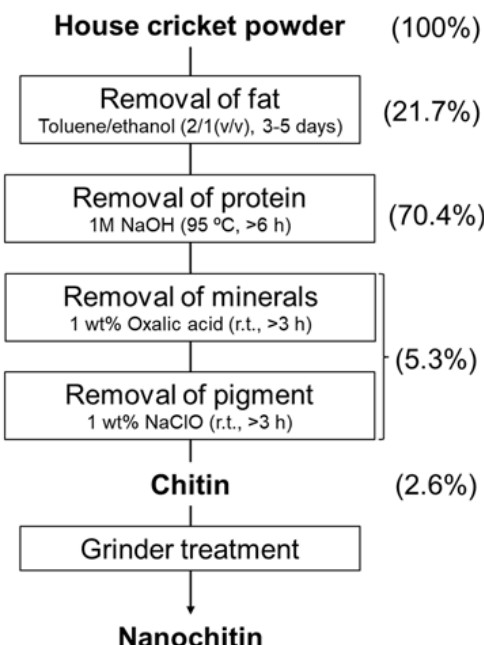

**Figure 1.** Preparation procedure for nanochitin from cricket and weight percentages of each extract.

*2.3. Preparation of Nanochitin from Cricket Chitin*

The cricket chitin was dispersed into 0.5 wt% acetic acid aqueous solution with 1.5 wt% concentration and passed through disk mill equipment (Super Masscolloider (MKZA12-20J) Masuko Sangyo, Kawaguchi, Japan) for mechanical treatment. The grinding stone clearance of the disk mill was set at $-0.15$ mm from the zero position (slight contact) of the two stones, and the rotational speed of the stone was 1500 rpm. After seven milling cycles, the dispersion was collected and stored in a refrigerator at 4 °C for further use. Crab shell nanochitin from snow crabs and red snow crabs was also produced in the same way as a sample for comparison.

*2.4. Characterization of Cricket Chitin and Nanochitin*

2.4.1. $^{13}$C CP/MAS-Nuclear Magnetic Resonance (NMR) Spectroscopy

Cricket chitin analysis was performed by $^{13}$C NMR spectrum with cross-polarization, and a magic-angle spinning technique was measured using a JNM-ECZ600R (JEOL Ltd., Tokyo, Japan). The spectrum was recorded at a $^{13}$C frequency of 150 MHz. The $^{13}$C spin-lattice relaxation time was 2 s. Cricket chitin powder was placed in a zirconia rotor and spun as fast as 5 kHz, and the contact time was 2 ms.

2.4.2. Fourier-Transform Infrared (FT-IR) Spectroscopy

The FT-IR spectrum of the cricket chitin was measured using an FT-IR spectrophotometer equipped with an ATR accessory (Spectrum 65, Perkin-Elmer Japan Ltd., Tokyo, Japan). The measurement was performed in the wavenumber range of 650 to 4000 cm$^{-1}$ with 32 scans at 4 cm$^{-1}$.

2.4.3. Scanning Electron Microscopy (SEM) Observation

The morphological structure of the cricket nanochitin was observed using an FE-SEM (JSM-6700F, JEOL Ltd.). The acceleration voltage was 2.0 kV. The dry sample was placed on carbon tape and coated with a platinum layer of about 2 nm by an automatic ion sputter coater at 10 mA for 90 s. To prepare samples for observation, ethanol was added to the nanochitin suspension and dried at 60 °C to avoid the aggregation of nanochitin.

### 2.4.4. Atomic Force Microscopy (AFM) Observation

AFM (Nanocute, SII Instruments, Chiba, Japan) was used to study surface morphology and measure the size of the nanochitin. It was diluted into 0.05 wt% by distilled water, dropped onto the surface of freshly cleaved mica substrate, and dried overnight at 40 °C. To investigate the size distribution, 336 nanochitin whiskers were randomly selected, and their diameters were measured.

### 2.4.5. Particle Size Measurement

The hydrodynamic diameter of the nanochitin dispersed in water was measured using the dynamic light scattering (DLS) method (ELSZ-1000ZS, Otsuka Electronics, Osaka, Japan). The concentration of the dispersion was 0.001 wt% to avoid coagulation. The pH was set to 3 to improve dispersion in water.

### 2.4.6. X-ray Diffraction (XRD) Profile

The crystalline structure of the nanochitin was investigated by XRD. The diffraction profile was detected using an X-ray generator (Ultima IV, Rigaku, Japan) at 40 kV in the goniometer scanning range of 5 to 45°. The degree of crystallization was calculated from the X-ray diffraction pattern using the following Equation (1) [18].

$$\text{Crystallinity index (\%)} = \frac{I_{110} - I_{am}}{I_{110}} \times 100 \tag{1}$$

where $I_{110}$ is the maximum intensity of diffraction of the peak at $2\theta = 22°$, and $I_{am}$ is the intensity of diffraction of the peak at $2\theta = 18°$.

The crystalline size was calculated from the full width of the peak at the half height of the source curve (FWTH) according to Scherrer's Equation (2) [19].

$$\text{Crystalline size (nm)} = \frac{K\lambda}{\beta_0 cos\theta} \tag{2}$$

where $K$ is constant (assumed to be 0.9), $\lambda$ is the wavelength of X ray-radiation (0.154 nm), $\beta_0$ (rad) is the width of the crystalline peak at half height, and $\theta(°)$ is the diffraction angle corresponding to the crystalline peak.

### 2.4.7. Ultraviolet-Visible (UV–Vis) Transmittance

The UV-Vis transmittance of the nanochitin (0.1%) was measured using a UV-Vis spectrophotometer (UV-2600i, Shimadzu, Kyoto, Japan). The transmittance spectra were measured with a wavelength range from 320 to 800 nm. The transmittance at 600 nm was measured periodically for 14 h to evaluate the dispersion stability.

### 2.4.8. Viscosity Measurement

The viscosity of the cricket nanochitin water dispersion (0.1 wt%) was measured using a Brookfield DV-E digital viscometer (Brookfield Engineering Laboratories, Inc., Middleboro, MA, USA) at 30 °C, which is close to room temperature. Spindle S63 was used in the measurement. The shear rate was 0.6 to 60 rpm. We took the viscosity after 5 min as the final value at all shear rates.

### 2.4.9. Specific Surface Area Measurement by Brunauer-Emmett-Teller (BET) Method

The BET method's specific surface area of the nanochitin was measured by $N_2$ adsorption at 77 K using an automatic Surface Area and Pore Size Distribution Measuring System (BELSORP mini II, MicrotracBEL Corp., Osaka, Japan). For measurements, aqueous nanochitin dispersions were replaced stepwise with methanol and *tert*-butyl alcohol, followed by freeze-drying.

### 2.4.10. Tensile Test of Cricket Nanochitin Sheet

The nanochitin suspension was vacuum filtered using a membrane filter (pore size: 0.45 mm). The obtained nanochitin sheets were hot-pressed at 4 MPa and 100 °C for 30 min to obtain a dried sheet with a diameter of 90 mm and a thickness of approximately 50 mm). The sheets were cut into a dumbbell shape (distance between grips: 33 mm, width of narrow section: 4 mm). A tensile test was performed using a universal testing instrument (ES-SX, Shimadzu) at a crosshead speed of 2.0 mm min$^{-1}$. At least five specimens were tested.

## 3. Results and Discussions

### 3.1. Isolation of Chitin from Crickets

Chemical treatments of house cricket powder were attempted to isolate chitin by following the method set out in Figure 1. The yield of the final product was 2.6%. The same process was attempted 10 times with good reproducibility of yields. The standard deviation of the yield was 0.9. The percentage of components extracted from crickets in each treatment was determined by the change in weight (Figure 1). The weight loss during the defatting and deproteinization processes was 21.7% and 70.4%, respectively. The results were reasonable because according to the nutritional information described on the house cricket product package (Futurenaut (Takasaki, Japan)), 100 g of the raw material contains 17.9 g of fat and 67.9 g of protein. From the protein assay using the Bradford method, it was confirmed that repeated treatment of the cricket with sodium hydroxide removed the protein to below the detection limit. Additionally, based on the percentage weight loss by thermogravimetric analysis, repeated treatment of cricket powder with oxalic acid reduced the amount of residual ash to less than 1% when the sample was heated to 700 °C. The degree of deacetylation of the chitin determined by electro conductivity titration [20] was 12.0%. It was higher than chitin from crab shells (6%). Since chitin is chemically bound to some proteins via amino groups, free amino groups are produced in the deproteinization process. This difference in the degree of deacetylation may be due to differences in the chemical structure of the chitin/protein complex. Actually, chitin molecules make covalent bondings with proteins to form proteoglycans [21].

The chemical structure of the purified crickets was evaluated by solid-state $^{13}$C NMR (Figure 2). The spectral data were in good agreement with those of chitin [22] and consisted of a clear signal derived from eight *N*-acetylglucosamine units. The signals located at chemical shifts at 173 ppm and 23 ppm are derived from the carbonyl and methyl groups of the acetyl group, respectively. Signals originating from carbons C1 to C6 of the glucosamine ring were observed in the range of 50 ppm to 110 ppm. The signals corresponding to C3 and C5 are clearly split at 76 and 74 ppm, suggesting that this is *a*-type chitin. On the other hand, *b*-type chitin is known to overlap those signals [23]. These results support the isolation of chitin from crickets.

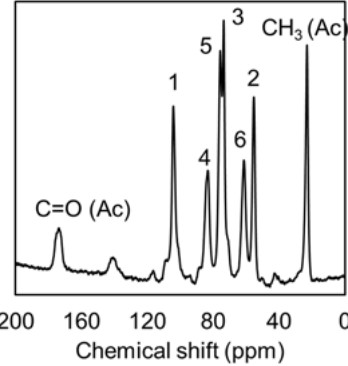

**Figure 2.** $^{13}$C NMR spectrum of cricket chitin.

FT-IR is one of the quickest and easiest techniques to prove the chemical structure of chitin. Figure 3 shows the FT-IR spectra of the products from cricket and crab shell chitin. The spectrum of the purified product from crickets was in perfect agreement with that of commercial chitin, indicating that chitin was isolated [24]. The peaks around 3420 cm$^{-1}$ and 3254 cm$^{-1}$ originate from O-H and N-H stretching vibrations. The peaks around 1620 cm$^{-1}$ and 1550 cm$^{-1}$ are attributed to amide I (C=O stretching vibrations) and amide II (N-H bending and C-N stretching vibrations), respectively. The peak of amide I was split, confirming that it is *a*-type chitin as well as chitin from crab shells. In *β*-chitin, on the other hand, it is known to appear as a single peak [23].

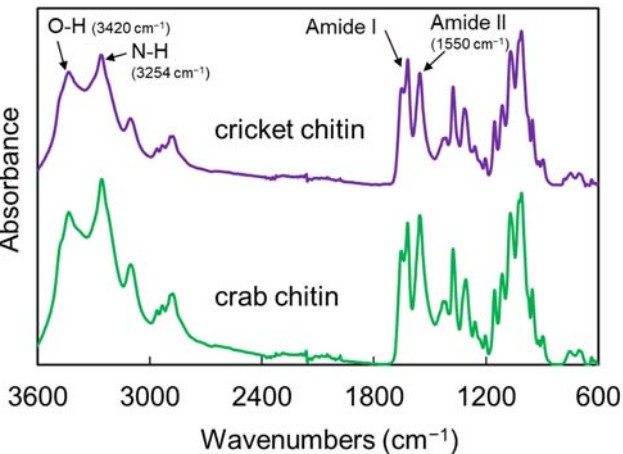

**Figure 3.** FT-IR spectra of cricket chitin crab chitin.

### 3.2. Preparation and Characterization of Cricket Nanochitin

Chitin isolated from cricket powder was ground in water using a disk mill. The milled chitin appeared as a white suspension. Its appearance was similar to that of the nanochitin derived from crab shells produced by conventional methods [2]. An SEM image of the milled product is shown in Figure 4 alongside conventional nanochitin from crab shells. The structure was a needle-like nanowhisker with a length of several hundred nm and a width of around 10 nm, indicating that nanochitin was obtained from the cricket powder [25]. Insect cuticles have a structure consisting of bundles of fibrils made of chitin. Their bundles are arranged in parallel to form a layered structure. Thus, this suggests that the organized chitin forming the cricket cuticle was disintegrated into chitin microfibrils by mechanical treatment. Nanochitin derived from crab shells obtained by conventional methods also has a whisker shape, but in comparison, the cricket nanochitin appears to be thicker and shorter. This may be due to differences in the morphology of the nanochitin origins that makes up the cuticle between insects and crustaceans.

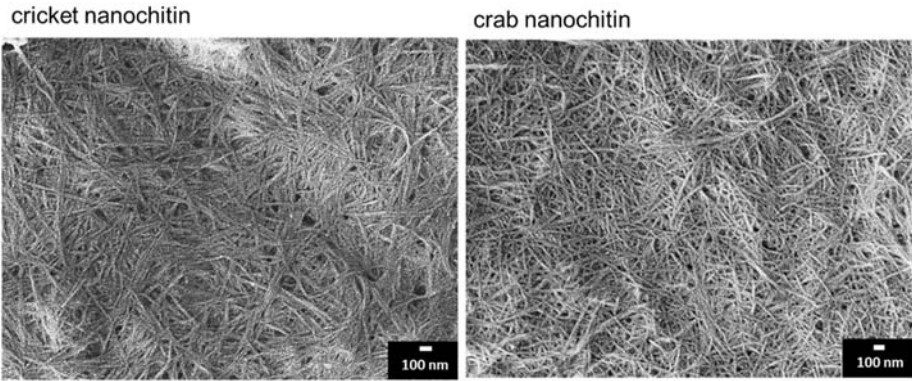

**Figure 4.** FE-SEM images of nanochitin from (**left**) cricket and (**right**) crab shell.

AFM observations were made to quantify the fiber width of nanochitin. The dilute nanochitin can be placed on a smooth mica substrate and scanned with a probe, and its height can be estimated as the fiber width. The fiber widths of more than 300 isolated nanochitin fibers were measured and their distribution was determined (Figure 5). Cricket nanochitin was observed with widths ranging from 1 nm to 23 nm. Moreover, 9–10 nm was the most common, with an average value of 10.1 nm. On the other hand, crab nanochitin was similarly observed with widths ranging from 1 nm to 23 nm. In addition, 4–5 nm was the most common, with an average fiber width of 7.46 nm. Thus, the results reflected the SEM observations that cricket nanochitin was thicker.

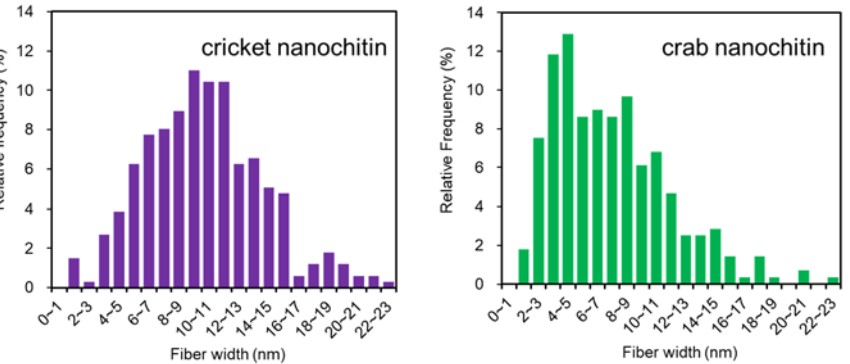

**Figure 5.** Distribution of length of nanochitin from (**left**) cricket and (**right**) crab shell.

Dynamic light scattering was used to measure the hydrodynamic diameter of nanochitin and its distribution (Figure 6). The fluctuations of nanoparticles in Brownian motion in solution provide information about the size of the nanochitin. The size distribution of cricket and crab nanochitin had two peaks each. The larger-sized peaks around 60,000 nm indicated coarse chitin particles with insufficient milling. The diameter of the coarse particles was smaller for cricket nanochitin. This suggests that cricket chitin may be more easily crushed. In the smaller peaks, cricket nanochitin was smaller in size compared to crab nanochitin. The average values were 522.5 nm and 928.1 nm, respectively. AFM observations suggest that the width of cricket nanochitin is greater than that of crab nanochitin, suggesting that cricket nanochitin is shorter than crab nanochitin. The result is consistent with the SEM observations. Chitin has a structure composed of microcrystals. The structure is then mechanically separated by milling and converted into nanochitin. Thus, the difference in morphology may be due to the different sizes of the chitin microcrystals in the exoskeletons of crickets and crabs.

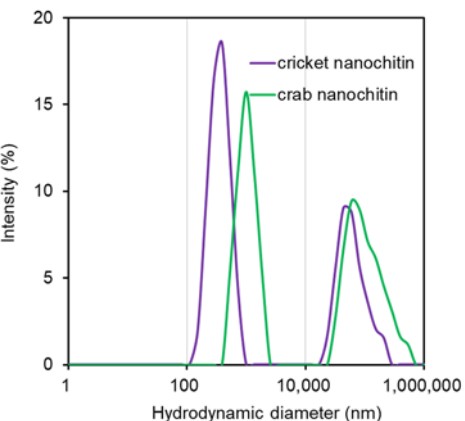

**Figure 6.** Hydrodynamic diameter of nanochitin from cricket and crab shell.

X-ray diffraction measurements were performed to analyze the crystal structure of cricket nanochitin (Figure 7). The reflection peaks at 9.1 and 19.1 degrees in the diffraction

pattern were identified as diffraction planes (020) and (110) of the orthorhombic crystal structure, respectively [19]. They were in good agreement with those from crab nanochitin, indicating that cricket nanochitin has an α-type antiparallel crystal structure. The relative crystallinity and crystallite size, which can be estimated from the XRD profiles, are shown in Table 1. The relative crystallinity of cricket nanochitin was 86.3%, a value as high as that from crab shells. It consists of microcrystals of chitin molecules arranged by hydrogen bonds. The crystallite sizes at the (020) and (110) planes were 6.1 nm and 4.3 nm, respectively, which were smaller than those of crab nanochitin, as determined from Scherrer's equation. This may be due to differences in the plane and angle of the chitin molecular chain during microfibril formation, as well as inter- and intramolecular hydrogen bonds in the crystal lattice.

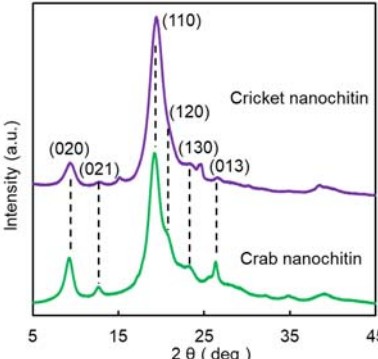

**Figure 7.** XRD profiles of nanochitin from cricket and crab shell.

**Table 1.** The relative crystalline index and crystalline size of cricket and crab nanochitin.

| Sample | Relative Crystalline Index (%) | Crystalline Size (nm) | |
|---|---|---|---|
| | | 020 | 110 |
| Cricket nanochitin | 86.3 | 6.1 | 4.3 |
| Crab nanochitin | 87.6 | 8.0 | 4.6 |

Differences in nanochitin morphology affect various physical properties of its aqueous dispersion, such as dispersibility and viscosity [26]. The UV-vis spectra of cricket and crab nanochitin aqueous dispersions and their transmittance over time are shown in Figure 8. In the spectra, the transmittance of cricket nanochitin was higher than that of crab nanochitin. For example, it was 14.7% higher at a shorter wavelength (400 nm), where it is more easily optically scattered on the nanochitin surface. This is due to differences in the shape of the nanochitin. The hydrodynamic diameter of cricket nanochitin is smaller than that of crab nanochitin. As a result, light scattering due to the difference in refractive index at the interface between nanochitin and water is less likely to occur, resulting in improved transparency [26]. Furthermore, in the case of cricket nanochitin, the hydrodynamic diameter of the poorly milled coarse particles is smaller than that of crab-derived particles, which also contributes to improved transparency. The degree of deacetylation is higher for cricket nanochitin than for crab chitin. The amount of amino groups on the nanochitin surface may also have an effect on transparency [27]. Next, the dispersion stability of nanochitin was evaluated. The transmittance of the aqueous dispersion of cricket nanochitin at 600 nm was about 50% and remained nearly constant for 14 h. This suggests that chitin nanowhiskers have a high affinity for water, forming a network and retaining water within their structure. Conventional chitin is basically undispersible in water [28], which limits its use. The high dispersibility and stability of cricket nanochitin in water facilitate the exploration of its functions and commercialization.

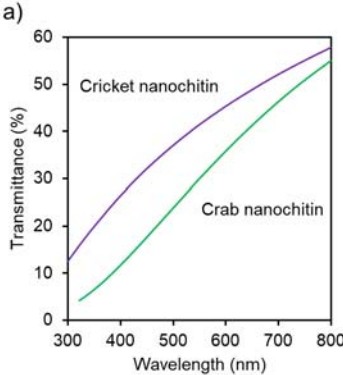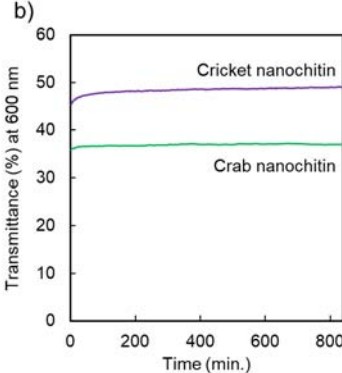

**Figure 8.** (**a**) UV-vis spectra and (**b**) transmittances at 600 nm of nanochitin from cricket and crab shell.

The viscosity of the nanochitin aqueous dispersion is shown in Figure 9. Cricket nanochitin exhibited so-called non-Newtonian fluid behavior, in which viscosity increased with increasing rotational speed (shear rate). This is due to the dispersion of whisker-shaped solid particles in water. Its viscosity was smaller than that of crab nanochitin. This is because cricket nanochitin is shorter than crab nanochitin [29]. When the fibers are shorter, the interaction associated with an entanglement between their particles is reduced, and thus the viscosity decreases. Nanochitin is highly viscous due to its morphology and high dispersibility in water. Cricket nanochitin with lower viscosity can be produced at higher concentrations, which is advantageous from an industrial standpoint.

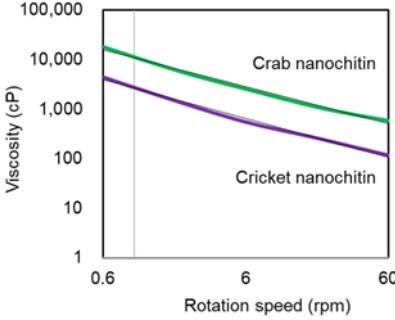

**Figure 9.** The viscosity of nanochitin dispersion from cricket and crab shell.

To determine the specific surface area of nanochitin, its aqueous dispersion was freeze-dried. If the aqueous dispersion is freeze-dried directly, the ice crystals coarsen, resulting in the agglomeration of the nanochitin. To avoid this, the water is gradually replaced with *tert*-BuOH before freeze-drying [30]. An SEM image of freeze-dried cricket nanochitin is shown in Figure 10. A regular layered structure consisting of nano-chitin was observed. This may represent the state of nanochitin in a solvent. Similarly to the lyotropic liquid crystals exhibited by rigid polymers, the whisker-shaped nanochitin crystals are assumed to be arranged by interactions between materials to form a more stable anisotropic structure in solvents. In fact, the cuticle of many insects has a spontaneously layered structure, with each layer forming a twisted cholesteric liquid crystal-like pattern. The specific surface area of its dried form was 742 $m^2$ $g^{-1}$, which was greater than that of crab nanochitin (523 $m^2$ $g^{-1}$). This may be because, according to the particle size distribution, cricket nanochitin has fewer coarse particles that are insufficiently ground. The value is large compared to the specific surface area of nanocellulose obtained by the mechanical milling of pulp. Nanocellulose, also a polysaccharide, is related to nanochitin in chemical structure and morphology. This difference may be due to differences in insect and wood structure. As noted above, insects have a relatively simple hierarchical structure, with nanochitin layers folded into a helical arrangement [25]. On the other hand, the higher-order structure

of the cell wall that makes up wood is quite complex. The wood cell wall has a hollow structure consisting of four layers, with nanocellulose microfibrils in each layer arranged in different directions, such as longitudinally, horizontally, and in a knitted structure [31]. Such so-called helical winding structures are the source of the tree's self-weight support. Such differences in the complexity of the higher-order structure may have affected the efficiency of milling, resulting in differences in the specific surface area.

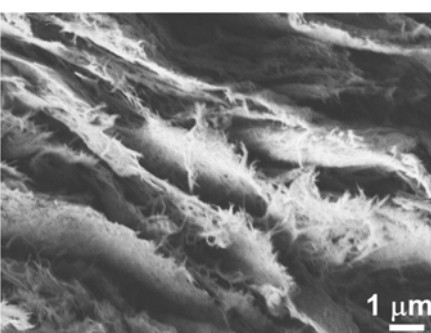

**Figure 10.** FE-SEM image of freeze-dried cricket nanochitin.

*3.3. Preparation and Mechanical Properties of Cricket Nanochitin Sheet*

Differences in nano-chitin morphology also affect the mechanical properties of processed products. Since nanochitin can be dispersed stably in water, the slurry can be dehydrated by filtration and hot pressing to form nanochitin sheets. The density and tensile test results of the obtained sheets are shown in Table 2. The density of the sheet was higher for cricket nanochitin than for that from crabs. This is because cricket nanochitin is shorter than crab. Whisker-shaped nanochitin is aligned horizontally to the plane of the nonwoven fabric by filtration but can be packed more densely when the fiber is short. Differences in density affect the Young's modulus and tensile strength. The elastic modulus of the cricket nanochitin sheet was 1.56 times greater than that of crab nanochitin, and the breaking strength was 1.12 times greater. On the other hand, the fracture strain was only 0.63 times as great. This is because when nanochitins are shorter, they have less opportunity to come into contact with other fibers. Then, in a tensile test, the sheet is more likely to fracture due to the concentration of stress in the sheet.

**Table 2.** Densities and mechanical properties of cricket and crab nanochitin sheet.

| | Density (g m$^{-3}$) | Young's Modulus (MPa) | Fracture Strength (MPa) | Fracture Strain (%) |
|---|---|---|---|---|
| Cricket nanochitin | 0.587 | 15.0 ($\pm$8.9) | 6.60 ($\pm$3.3) | 0.39 ($\pm$0.05) |
| Crab nanochitin | 0.453 | 9.6 ($\pm$3.9) | 5.89 ($\pm$2.3) | 0.62 ($\pm$0.03) |

The numbers in parentheses mean errors.

**4. Conclusions**

First, chitin can be isolated from crickets. A series of extraction processes were carried out to sufficiently remove proteins and minerals. The resulting chitin had an $\alpha$-type crystal structure, and the degree of deacetylation was higher than that of crab chitin. Then a grinding process was able to convert it into whisker-shape nano-chitin. The width of the fiber was about 10 nm, which was thicker than crab nanochitin, while the hydrodynamic diameter was smaller. Nanochitin can be dispersed stably in water. It is easy to process and formulate, which is advantageous in commercialization and in evaluating functionality too. Differences in shape affect the physical properties. Cricket nanochitin dispersion is more transparent and less viscous than crab nanochitin. Moreover, the dry sheet was higher in density, tensile strength, and elasticity. Crickets have been added as a new source of nanochitin. Although the proportion of protein is higher than that of crab shells, the yield would be improved if the inedible part, the hull, were selectively used as a raw material.

With the spread of insect diet as a protein source with low environmental load, if the technology to efficiently separate the outer skin from the inner protein is developed, it is expected to be utilized as a raw material for nanochitin.

**Author Contributions:** K.K. conducted all experiments and analytical characterization. T.M. conducted analytical characterization. S.I. and H.I. conceived the presented idea and designed the research. K.K. and S.I. contributed to writing the manuscript. All authors have read and agreed to the published version of the manuscript.

**Funding:** This research received no external funding.

**Institutional Review Board Statement:** Not applicable.

**Informed Consent Statement:** Not applicable.

**Conflicts of Interest:** The authors declare no conflict of interest.

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
