# Peer review of "Preparation of Nanochitin from Crickets and Comparison with That from Crab Shells"

_jcs, doi:10.3390/jcs6100280_

Round 1
Reviewer 1 Report
The reviewed work presents the results of research on the production of nanochitin from crickets. The authors obtained chitin by extraction method, which was then prepared in a nanometric form by grinding.
The authors characterized the produced material and compared the obtained results with nanochitin from crab shells.
The theoretical part of the work is properly prepared, however, there is no clear indication of the novelty of the work.
Methods of material characterization were selected appropriately.
In my opinion, the reviewed manuscript should be divided into two separate research papers on the characteristics of the obtained nanochitin, and another on the creation of thin films and the characteristics of their properties, e.g. mechanical and structural properties.
In the section on mechanical properties in regard to nanochityn sheet preparation, the pressing conditions (pressure, sample size, layer thickness, concentration from which it was obtained) are missing. The sample dimensions are missing. Basic statistical analysis is missing in the item on the description of mechanical properties. Therefore, I suggest excluding these results, developing them further, and preparing another manuscript.
In Fig. 4, what is the point of including a photo of the sample vial? This disturbs the perception of the presented structure. The method of drying such a sample may have an influence on the obtained structure, please complete the information on preparation
Is the structure shown in Fig. 10 a sample breakthrough?
Author Response
Dear reviewer #1:
Thank you very much for your helpful comments. We carefully considered reviewers’ comments and corrected. Thanks to the reviewers' qualified advice, the revised version is better for readers. We believe that the revised manuscript is acceptable to the Journal of Composites Science.
The reviewed work presents the results of research on the production of nanochitin from crickets. The authors obtained chitin by extraction method, which was then prepared in a nanometric form by grinding.
The authors characterized the produced material and compared the obtained results with nanochitin from crab shells.
The theoretical part of the work is properly prepared, however, there is no clear indication of the novelty of the work.
Methods of material characterization were selected appropriately.
In my opinion, the reviewed manuscript should be divided into two separate research papers on the characteristics of the obtained nanochitin, and another on the creation of thin films and the characteristics of their properties, e.g. mechanical and structural properties.
The results on "Preparation and characterization of cricket nanochitin" and "Preparation and mechanical properties of cricket nanochitin sheet" are divided into separate sections (section 3.2. and section 3.3.).
In the section on mechanical properties in regard to nanochityn sheet preparation, the pressing conditions (pressure, sample size, layer thickness, concentration from which it was obtained) are missing. The sample dimensions are missing. Basic statistical analysis is missing in the item on the description of mechanical properties. Therefore, I suggest excluding these results, developing them further, and preparing another manuscript.
Pressing conditions for making sheets were described in experimental section 2.4.10 (Line 166-172). The errors in the mechanical properties of the sheets were also listed in Table 2.
In Fig. 4, what is the point of including a photo of the sample vial? This disturbs the perception of the presented structure. The method of drying such a sample may have an influence on the obtained structure, please complete the information on preparation
Photographs of the appearance of nanochitin dispersion were excluded from Fig. 4. Sample preparation for SEM observation was described in Page 3, Line 123-124.
Reviewer 2 Report
The comments are in attached file.

Author Response
Dear reviewer #2:
Thank you very much for your helpful comments. We carefully considered reviewers’ comments and corrected. Thanks to the reviewers' qualified advice, the revised version is better for readers.We believe that the revised manuscript is acceptable to the Journal of Composites Science.
Reviewer #2:
The introduction should be written as state -of -the -art with added literature. As it is written now, there is confused since the reader do not understand what was done within this study and what was already done by other authors (as cited reference). The state of the art should contain the info regarding nanochitin production, structure that resulted in properties – advantages and disadvantages, etc., accompanied with literature. By my opinion, the manuscript should be re-written to be oriented toward “composite” science, otherwise I do not see any added value for this journal.
P1 lines 26-33: the reference should be added
P 1, lines 26-29: Research paper on an overview of chitin was cited as Ref. 1.
P1 line 35: The reference 12: Is these the result of this study?
P1 lines 33-35: To avoid misunderstanding, the sentence has been corrected to the passive voice.
P1 line 40: which mechanical properties?
P1 line 42: Specific words regarding mechanical properties were described.
P1 line 45: references11, 12 Is these the result of this study? You stated “we” then the lit is added? It is rather confusing, thus needs to be clarified
P2 line2 46-47: To avoid confusion, I have changed that sentence to the passive voice.
P2 lines 47-48: what properties – how and where …within health food, care, etc.
P2 line2 50-52: Specific characteristics of nanochitin for commercialization were described.
P2 lines 53-56: how the “food” is connected to composite science? Needs to be re-written, to be focused toward composite science.
P2 lines 53-57: The sentence is intended to use some of the crickets produced as an alternative source of protein as a chitin source.
P2 line 64 what differences? Needs to be highlighted
P2 line 68: The words which the DIFFERENCES indicate are specifically described.
P2 line 70: check the grammar
P2 line 75: Grammatical error has been corrected.
P2 line 71: what other chemicals? Needs to be listed
P2 lines 76-77: Chemicals used in this study were listed.
P2 line 80: Bradford method – reference source should be added
P2 line 85: I cited a paper describing the operation of the Bradford method as Ref. [17].
P2 line 81: water – ultra pure, distilled; etc.?
P2 line 86: Distilled water was used in this study.
P2 line 81: mineral – singular or plural? Which mineral – need to be listed
P2 line 87: Mineral refers to all inorganic constituents included in crickets.
P2 line 84: filtrated – how? Size filtration paper?
P2 line 86: Detailed filtration conditions are described.
P3 line 99: where both of nanochitin sources (i.e., crab compared to cricket characterised) it is not so obvious thus needs to be explained
P3 lines 104-105: Information about the type of crab shells were described.
P4 line 12: why used so low concentration? Why used Ph=3 -needs to be explained
P4 lines 134-135: The reason for the concentration and pH of the sample for the DLS measurement is described.
P4 lines 131 and 136: both equations used should be numbered
P4 lines 141 and 146: Numbers were provided to the two equations (1) and (2).
P 4 line 148: 2.4.8. viscosity – why used temperature of 30 degrees – needs to be explained
P4 line 158: The reason for measuring viscosity at 30°C was described.
P4 line 166 - 3.1 isolation from crickets: good reproducibility – how much – needs to be added
P5 lines 178-179: Errors of the yield were described to show reproducibility.
P5 line 170: the name of product and the name of producer should be added
P5 line 183: The name of the company was described.
P5 line 177: the reference 19 – what I the propose of added reference – is 12% the result obtain in this study or taken from literature 19 -this needs to be clarified
P5 line 189: The position of the citation number [20] has been changed to avoid misunderstanding.
P5 line 180: more detailed explanation of differences in structure is needed
P5 lines 193-194: The differences in structure were explained in more detail and relevant paper was cited as Ref. 21.
P5 line 182: in good agreement with chitin – it should be listed in Fig 2 for comparison – this needs to be added and re-written
P5 line 196: Relevant literature was cited for comparison of 13C NMR spectrum.
P5 line 200. Single peak -at which cm-1 – needs to be also indicated in Fig 3
Figure 3: Wavenumbers of each single peaks were listed in Fig. 3.
P6 line 207 – in Fig 4 – both products are shown?
Page 6 line 222: Conventional nanochitin derived from crab shells was also described in the text.
P6 lines 207-209: why the 2 sentences are both with thus – is there any connection? needs to be clarified and rewritten
P6 line 224: "Thus" was deleted and then revised the text.
P6 line2 214-215: differences in structure should be explained in more details
P6 lines 230-231: The text about differences in structure was revied to be more specific.
P6 line 226-227: this is not correct, since the above sentence by SEM stated that cricket fibres are shorter and thicker – needs to be clarified
P7 line 243-244: Both SEM and AFM images show that cricket nanochitin is thicker than crab shell nanochitin.
P8 line 264: which physical properties – needs to be listed along with cited reference
P8 lines 281-282: We have specifically described what the physical properties indicate. We also cited relevant literature.
P8 line 280-281: undispersible…limit use - some reference should be added
P8 line 299: Relevant papers were cited.
P9 line 312 and line 314: what wood structure – how is this relevant – need to be clarified
Page 10 lines 328-329: It was noted that nanocellulose is a related substance to nanochitin.
P10 line 315: the wood – how Is relevant to chitin/cricket/crab– needs to be clarified
Page 10 lines 330-333: It was noted that wood cell wall and crickets differ in the complexity of their higher order (hierarchical) structures.
P10 line 325: non-woven fabric suddenly appears – how it is now relevant for the study – because it was not mentioned before within the whole text of manuscript. This should be clarified while using it the whole text must be re-written…thus some specification of non-woven (type, structure, producer, etc.) should be added and re-write the aim of the study why it is relevant, etc.
Page 11 lines 345, 346,, and 347: To avoid confusion among readers, we have changed the term "non-woven fabric" and unified it with “sheet”.
P10 line 332-333: check the grammar -it is not understandable
Page 11 lines 355-357: The explanation of why the fracture strain decreased was not clear and has been corrected.
Yours sincerely,
Round 2
Reviewer 1 Report
The authors have made additions to the manuscript, which significantly increase the scientific value of the work. In my opinion, the revised manuscript meets the requirements for papers suitable for the J. compos. Sci., and therefore I ask the editors to admit the manuscript for further publication stages.